# Multidimensional Statistical Technique for Interpreting the Spontaneous Breakthrough Cancer Pain Phenomenon. A Secondary Analysis from the IOPS-MS Study

**DOI:** 10.3390/cancers13164018

**Published:** 2021-08-10

**Authors:** Marco Cascella, Anna Crispo, Gennaro Esposito, Cira Antonietta Forte, Sergio Coluccia, Giuseppe Porciello, Alfonso Amore, Sabrina Bimonte, Sebastiano Mercadante, Augusto Caraceni, Massimo Mammucari, Paolo Marchetti, Rocco Domenico Mediati, Silvia Natoli, Giuseppe Tonini, Arturo Cuomo

**Affiliations:** 1Division of Anesthesia and Pain Medicine, Istituto Nazionale Tumori, IRCCS Fondazione G. Pascale, 80100 Naples, Italy; m.cascella@istitutotumori.na.it (M.C.); gennaro.esposito@istitutotumori.na.it (G.E.); c.forte@istitutotumori.na.it (C.A.F.); s.bimonte@istitutotumori.na.it (S.B.); a.cuomo@istitutotumori.na.it (A.C.); 2Epidemiology and Biostatistics Unit, Istituto Nazionale Tumori, IRCCS Fondazione G. Pascale, 80100 Naples, Italy; sergio.coluccia@hotmail.it (S.C.); g.porciello@istitutotumori.na.it (G.P.); 3Melanoma and Sarcoma Surgery Unit, Istituto Nazionale Tumori, IRCCS Fondazione G. Pascale, 80100 Naples, Italy; a.amore@istitutotumori.na.it; 4Anesthesia and Intensive Care & Pain Relief and Supportive Care, 00185 La Maddalena, Italy; terapiadeldolore@lamaddalenanet.it; 5Palliative Care, Pain Therapy and Rehabilitation, National Cancer Institute, IRCCS Foundation, 20133 Milan, Italy; augusto.caraceni@istitutotumori.mi.it; 6Primary Care Unit, ASL RM1, 00193 Rome, Italy; massimo.mammucari@libero.it; 7Department of Clinical and Molecular Medicine, La Sapienza University of Rome, 00185 Rome, Italy; paolo.marchetti@uniroma1.it; 8Palliative Care and Pain Therapy Unit, Careggi Hospital, 50139 Florence, Italy; mediatir@aou-careggi.toscana.it; 9Department of Clinical Science and Translational Medicine, University of Rome Tor Vergata, 00133 Rome, Italy; silvia.natoli@uniroma2.it; 10Medical Oncology Department, Campus Bio-Medico University of Rome, 00128 Rome, Italy; g.tonini@unicampus.it

**Keywords:** cancer pain, breakthrough cancer pain, cluster analysis

## Abstract

**Simple Summary:**

Pain is one of the most common and debilitating symptoms in cancer patients. A clinical peculiarity of cancer pain is the breakthrough cancer pain (BTcP), which is defined as a temporary exacerbation of pain that “breaks through” a phase of adequate pain control by an opioid-based therapy. The NP-BTcP occurs in the absence of any specific activity. In this paper, we addressed the topic through a mathematical approach to provide many indications for identifying the diagnostic and therapeutic gaps in NP-BTcP management.

**Abstract:**

Breakthrough cancer pain (BTcP) is a temporary exacerbation of pain that “breaks through” a phase of adequate pain control by an opioid-based therapy. The non-predictable BTcP (NP-BTcP) is a subtype of BTcP that occurs in the absence of any specific activity. Since NP-BTcP has an important clinical impact, this analysis is aimed at characterizing the NP-BTcP phenomenon through a multidimensional statistical technique. This is a secondary analysis based on the Italian Oncologic Pain multiSetting—Multicentric Survey (IOPS-MS). A correlation analysis was performed to characterize the NP-BTcP profile about its intensity, number of episodes per day, and type. The multiple correspondence analysis (MCA) determined the identification of four groups (phenotypes). A univariate analysis was performed to assess differences between the four phenotypes and selected covariates. The four phenotypes represent the hierarchical classification according to the status of NP-BTcP: from the best (phenotype 1) to the worst (phenotype 4). The univariate analysis found a significant association between the onset time >10 min in the phenotype 1 (37.3%)’ vs. the onset > 10 min in phenotype 4 (25.8%) (*p* < 0.001). Phenotype 1 was characterized by the gastrointestinal type of cancer (26.4%) with respect to phenotype 4, where the most frequent cancer affected the lung (28.8%) (*p* < 0.001). Phenotype 4 was mainly managed with rapid-onset opioids, while in phenotype 1, many patients were treated with oral, subcutaneous, or intravenous morphine (56.4% and 44.4%, respectively; *p* = 0.008). The ability to characterize NP-BTcP can offer enormous benefits for the management of this serious aspect of cancer pain. Although requiring validation, this strategy can provide many indications for identifying the diagnostic and therapeutic gaps in NP-BTcP management.

## 1. Introduction

Despite the many therapeutic possibilities, pain remains one of the most common and debilitating symptoms in cancer patients. Its prevalence is approximately 55% during anticancer treatment, 40% after therapy, and about 66% in the advanced and progressive phase of disease stages [1,2]. Although the numerical data are useful for obtaining a quantitative estimate of the problem, the precise characterization of this complex symptom is mandatory. This involves defining the various clinical presentations.

In the 11th revision of the International Classification of Diseases (ICD), a task force of the International Association for the Study of Pain (IASP) has provided a new classification of chronic pain [3]. It distinguishes between chronic primary pain and chronic secondary pain conditions. Chronic cancer pain is a subtype of chronic secondary pain, which is defined as chronic pain caused by primary cancer itself or metastases or its treatment [4]. A clinical peculiarity of cancer pain is the breakthrough cancer pain (BTcP). It is a temporary exacerbation of pain that “breaks through” a phase of adequate pain control by an opioid-based therapy [5,6]. Notably, BTcP can affect up to 70% of cancer patients [7,8]. Clinically, an episode of BTcP is of short duration (15–30 min), moderate-to-severe intensity, and short latency between onset and peak of intensity (a few minutes) [9,10]. Its clinical presentations, however, may vary depending on multiple factors, such as the clinical setting and type and degree of the background pain [11,12].

BTcP is divided into two categories, including the predictable and the non-predictable BTcP (NP-BTcP) type. The predictable, or incident type, is further categorized into three subclasses, including the volitional BTcP (caused by a voluntary act), the non-volitional subtype (caused by an involuntary act), and the procedural pain. The NP-BTcP occurs in the absence of any specific activity. It is also indicated as spontaneous or idiopathic BTcP [5,11].

The clinical and therapeutic implications, and probably the pathophysiology, are very different. For example, since the predictable BTcP is usually triggered by voluntary (e.g., dressing changes) and not voluntary movements (e.g., bowel spasm) or therapeutic procedures, adequate knowledge of the phenomenon can facilitate the prevention and implementation of effective therapeutic strategies. On the other hand, since it is not possible to establish its triggering events, the clinical and therapeutic implications of NP-BTcP are of fundamental importance. The idiopathic form has many aspects yet to be defined. However, since it accounts for about 70% of the total cases of BTcP [5] and is in nature unpredictable, with little or no warning, it should be carefully addressed.

The study of the phenomenon can be performed through clinical investigations. Another strategy is the analysis of high sample size datasets using ad hoc statistical methods. On these premises, this work aims to characterize the NP-BTcP phenomenon through a multidimensional method applied to the most copious database collected so far.

## 2. Materials and Methods

### 2.1. The IOPS-MS Dataset

This is a secondary analysis based on the Italian Oncologic Pain multiSetting—Multicentric Survey (IOPS-MS) that was carried out for dissecting the BTcP phenomenon [5,7]. Details concerning enrollment of patients and all recorded variables are described elsewhere [5,7]. The local Ethical Committee approved the protocol, and each patient signed informed consent.

Inclusion criteria were:Age greater than 18 years;Diagnosis of cancer at any stage; well-controlled and stable background pain with an intensity ≤ 4 (on a 0–10 numerical rating scale, NRS);Presence of BTcP episode of moderate–severe intensity, clearly distinguished from background pain.

Exclusion criteria were:No cancer diagnosis;Unstable and/or uncontrolled background pain (>4/10);No relevant peaks in pain intensity (<5/10);Poor collaboration or refusal to participate.

Considering that in spontaneous BTcP, more than 3–4 episodes per day usually indicate uncontrolled background pain (requiring a careful optimization of basal pain) [5,6,7,11,12], we adopted the cut-off of 4 episodes per day.

Among all recorded variables from the original study, some variables were selected for this secondary analysis: age, gender, setting, type of primary tumor, onset, type of BTcP pain, Karnofsky status, and type of physician were recorded. Patients were asked about the average time of meaningful pain relief after their BTcP medication (≤10 min, >10 min). The study was observational, and pharmaceutical therapies were used according to local policy, without following strict protocols.

In the original IOPS-MS dataset, *n* = 4016 cases were entered. Of these, 30.5% (*n* = 1225) were incident BTcP and 69.5% (*n* = 2791) NP-BTcP [7]. The statistical analysis of this study concerned the latter subgroup.

### 2.2. Statistical Analysis

Descriptive statistics were provided for BTcP characteristics and main variables. At first, to characterize the BTcP profile, a Pearson correlation was performed among the following quantitative discrete/continuous variables:(a)BTcP intensity score (NRS 5–10) (quantitative discrete variable).(b)Number of BTcP episodes per day (1–4) (quantitative discrete variable).(c)Duration of BTcP (quantitative continuous variable).

By selecting the quantitative discrete variables with the highest correlation (*a* and *b* variables; *p* < 0.001) and by adding the categorical variable “BTcP type” (nociceptive, neuropathic, both), a multidimensional analysis was performed. The multiple correspondence analysis (MCA) determined the identification of groups (termed as ‘phenotypes’). The MCA plot showed the theoretical clusters, and a further confirmative hierarchical clustering principal components analysis (HCPC) [13] was implemented to compare the theoretical findings through an automatic classification criterion [14]. It is a technique that reduces the number of observations by classifying them into homogeneous clusters, identifying the groups without previously knowing group memberships or the number of possible groups. Moreover, the Elbow plot is a graphical tool that helps to choose the best number of clusters for HCPC. It uses k-means to calculate WSS (within groups sum of squares) and suggests the best representation of clusters. Maximization of the inertia between centroids and barycenter (Ward method) was adopted in the HCPC function to classify the groups.

Finally, a univariate analysis was performed to observe differences between the 4 phenotypes; a chi-square test was used to measure categorical variables, and a *p*-value less than 0.05 was considered statistically significant (Figure 1). The statistical analysis was carried out using R (version 4.0.2) software by using the FactoMineR package [15] and SPSS version 26 [16].

## 3. Results

Among patients with NP-BTcP (*n* = 2791) from the original study, 120 presented more than four episodes per day. Those patients were excluded from this secondary analysis; thus, 95.7% of the total set was analyzed (*n* = 2671).

Patients’ characteristics are described in Table 1. Fifty-five percent of patients were male, and over 52% were above 65 years of age; the most frequent setting was the day-hospital setting (47%), and the most frequent primary tumor was the lung carcinoma (25%). The number, intensity, and type of BTcP were described; about 68% of the onset was less than 10 min, and regarding the therapy, 49% of patients received rapid-onset opioids (ROOs) for BTcP management.

MCA finding was shown in Figure 2. The four clusters (phenotypes) represent a theoretical classification according to the BTcP status, which from P1 (phenotype 1) to P4 (phenotype 4) gradually get worse.

An automatic procedure recognizes the six theoretical clusters groups (within groups sum of squares 6, WSS_6_ = 509.13). The k-means method, represented by the Elbow plot (Figure 3), suggests that a good representation of clusters is given from 4 to 10 groups. Therefore, a parsimonious point of view was adopted, and four clusters final groups (phenotypes) were selected (WSS_4_ = 838.93). According to cluster estimation, the agreement between induced classification and automatic classification was 31.6%.

The features of the four clusters/phenotypes are synthesized in Figure 4.

The univariate analysis shows the associations between the four phenotypes and selected variables (Table 2). A borderline significance was observed for age: young patients (<55 years) were more favorable to be in P4 (24.2%) compared with the elderly (≥75 years) who were allocated in P1 (25.9%) (*p* = 0.055). A significant association was assessed between the onset time >10 min in phenotype 1 (37.3%) vs. the onset >10 min in phenotype 4 (25.8%) (*p* < 0.001); inpatients were mainly allocated into cluster P4, while those in palliative care settings who expressed phenotype 1 (43.6% vs. 26.6%, respectively *p* < 0.001). About cancer type, P1 was characterized by gastrointestinal (26.4%) respect to P4, where the most frequent cancer affected was the lung (28.8%) (*p* < 0.001).

Phenotype 4 was mainly managed with rapid-onset opioids (ROOs), while in phenotype 1 many patients were treated with oral, subcutaneous, or intravenous morphine (56.4% and 44.4%, respectively; *p* = 0.008). Moreover, the number of patients who did not receive therapy (ROOs, morphine, and other therapy) decreased from P1 to P4 (16.5% and 8.1%, respectively; *p* = 0.003) (Figure 5).

## 4. Discussion

To our knowledge, this is the first study aimed at characterizing the NP-BTcP phenomenon. Several pieces of evidence showed that BTcP cannot be intended as a single nosological entity but includes different and changing pathogenetic mechanisms justifying its sub-typing [7,9,17]. Consequently, methods useful for dissecting the NP-BTcP forms can offer great help for clinicians and important benefits for patients [18].

The main challenge for BTcP management is the implementation of new procedures that can be useful for both diagnostic and therapeutic approaches. Recently, Pantano et al. [19] performed a machine learning approach to identify possible subgroups of both types of BTcP. Their algorithm was based on BTcP therapy satisfaction, clinical features, basal pain, and rapid-onset opioids. This approach allowed the identification of 12 distinct BTcP clusters. In our analysis, through the automatic classification method, four clusters of NP-BTcP were identified. The clusters were obtained by combining two continuous variables (BTcP intensity and number of BTcP episodes per day) with the categorical variable BTcP type (nociceptive, neuropathic, or both) (Figure 2).

Subsequently, the characteristics assumed by the individuals in each group were projected. In turn, a rule of classification was developed (Figure 4). This approach could facilitate rapid identification of the patient who is potentially at higher risk of developing more severe forms of NP-BTcP in terms of pain intensity, type, and number of episodes per day. Notably, although the method initially suggested a classification of six groups, the choice was given to the four groups, given the best goodness of estimate (WSS_4_ = 838.93). For all these reasons, the goodness of fit was 32%, which represents the explained variation of the entire phenomenon. Furthermore, it must be considered that the model was based on three variables and large sample size. According to the hierarchical order assumed by the phenotypes, it was easier to recognize the best group (P1) versus the worst one (P4).

The data emerging from the categorization indicate that P1 expresses nociceptive pain. Although even four episodes per day can occur, the NRS (0–10) is of intensity between 5 and 6. On the other hand, P4 is characterized by episodes of neuropathic pain or both pains of severe intensity. The intermediate clusters (P3 and P4) may have a nociceptive type of NP-BTcP but of high intensity, or they express neuropathic pain/both pains of submaximal intensity.

Subsequently, the univariate analysis showed that phenotype 1 is characterized by older age, slow onset, gastrointestinal as primary tumor, and greater propensity to be treated in the context of the palliative care setting. On the contrary, the main features of phenotype 4 are younger age and rapid onset; furthermore, it most frequently concerns inpatients affected by lung cancer. Moreover, regarding the NP-BTcP therapy, the worst phenotype (P4) was mainly managed with ROOs; on the contrary, in phenotype 1 many patients were treated with oral, subcutaneous, or intravenous morphine. Interestingly, guidelines on cancer pain treatment do not refer to the severity of BTcP and provide different guidance regarding its treatment. For instance, the European Society for Medical Oncology (ESMO) guidelines recommended the use of fentanyl formulations for the treatment of BTcP (level of evidence I, degree of recommendation A). In the same guidelines, morphine received a level of evidence of II and a degree of recommendation of B [20]. On the other side, despite the WHO guidelines stated that BTcP should always be relieved with rescue medicine based on clinical experience and patient need, immediate-release or slow-release morphine was recommended [21]. Nevertheless, generic guidelines continue to suggest the use of oral opioids, whereas specific BTcP guidelines recommend the use of ROOs as rescue medication. According to Davis et al. [6], the different attitudes towards BTcP management do not reflect research evidence but personal opinions.

A great concern is the lack of treatment of NP-BTcP for each phenotype. Notably, about 17% of patients in P1 did not receive any treatment and, even if this percentage decreases up to P4, there is a statistically significant difference compared with those who received ROOs, morphine, and all other therapies. Despite the guidelines on the topic, proper cancer pain treatment remains a paramount unmet medical need for cancer patients [22,23].

The ability to characterize NP-BTcP can offer enormous benefits for the management of this serious aspect of cancer pain [24], facilitating personalized treatment of cancer pain [25,26]. This strategy can provide many indications for identifying the diagnostic and therapeutic gaps in NP-BTcP management [27,28].

Particular care perspectives are very interesting. The possibility of developing a mobile application to quickly allocate the patient in one of the four clusters is one of these perspectives. This method could implement systems already in use or being tested. The Pain Assessment in Cancer Patients by Machine Learning (PASCALE) study (NCT04726228) is ongoing; it is aimed at evaluating the possibility that telemedicine can improve the quality of life of cancer patients through self-management and remote monitoring solutions. The study involves the development of an application that also records information on the BTcP features. After the first phase of implementation of the database and machine learning for identifying objective characters of pain, the authors plan to include the four clusters to study multiple features of NP-BTcP (clinical data, therapy, impact on quality of life, and others).

In addition to experimental approaches, telemedicine tools could be used to assist those patients who are immediately allocated to the worst clusters (P3 and P4). For these patients, the management of NP-BTcP must be very aggressive to counteract its impact on the quality of life.

If adequate resources are available, even the so-called best phenotypes (P1 and P2) can benefit from telemedicine. The experience accumulated in the management of patients with cancer pain during the pandemic suggests that many episodes of NP-BTcP are not reported during periodic consultations but are recorded by the patient in the monitoring system [29]. In other words, the impact on quality of life is always present, but both patients and clinicians tend to underestimate the effect [30,31].

This study has multiple limitations. The most important limitation concerns the assumption that it is a secondary analysis based on a survey. However, our starting point remains the most abundant dataset on the subject. Moreover, the original IOPS study was not drawn to perform this analysis. This gap could be overcome by the development of prospective studies to verify the reliability (validation) of the proposed classification system (Table 2). Another limitation regards the mathematical approach as the method works only with complete data. Nevertheless, we adopted a dataset with no or few missing values.

## 5. Conclusions

The idiopathic form of BTcP has many aspects yet to be defined, and the characterization of the phenomenon means being able to improve its management. In-depth knowledge is also mandatory because NP-BTcP accounts for over two thirds of all BTcP cases. While waiting to obtain data from prospective studies, dataset analyzes carried out according to ad hoc statistical approaches can allow us to define clusters (phenotypes) that differ in the type of pain, intensity, and number of daily episodes. It can be deduced, for instance, that those suffering from lung cancer may be classified in the worst phenotype of NP-BTcP (P4), and this prerequisite presupposes greater diagnostic attention. Finally, a more precise definition of clusters could lead to important therapeutic perspectives. For example, a mobile application could be useful for the clinician to frame individual patients. To be more precise, patients who fall into the worst phenotypes could be followed up under telemedicine programs. The aim is to provide tailored management of the NP-BTcP through designing prospective studies for the validation of this multidimensional statistical approach.

## Figures and Tables

**Figure 1 cancers-13-04018-f001:**
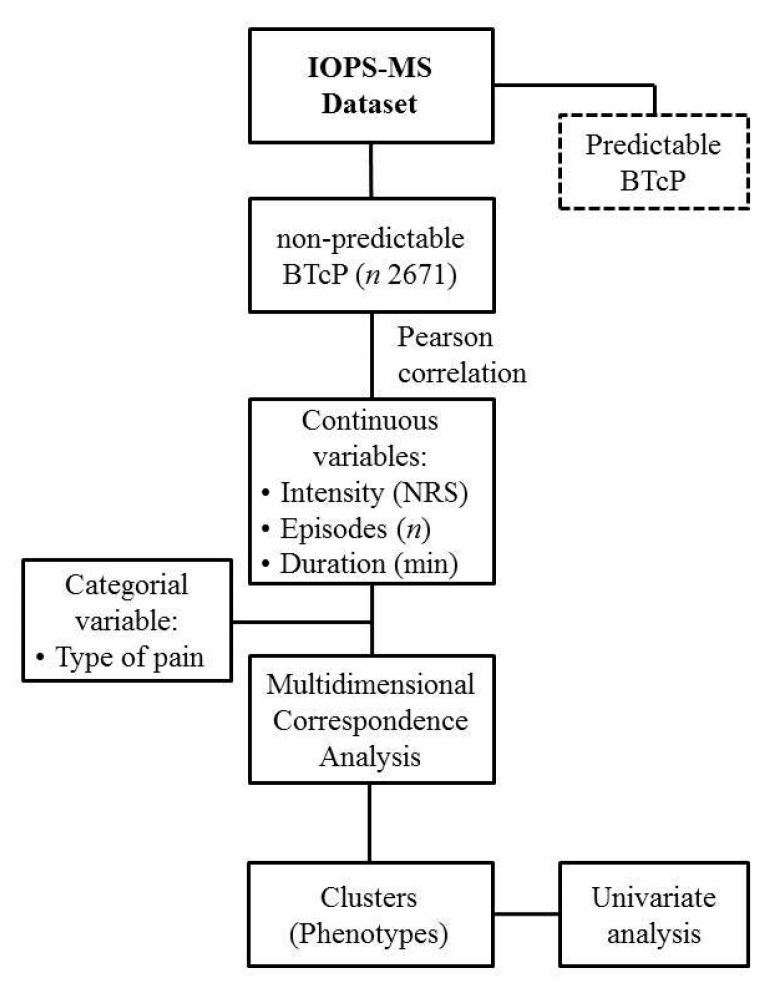
Flow chart. Abbreviations. IOPS—Italian Oncological Pain Survey; BTcP—breakthrough cancer pain; NRS—numerical rating scale.

**Figure 2 cancers-13-04018-f002:**
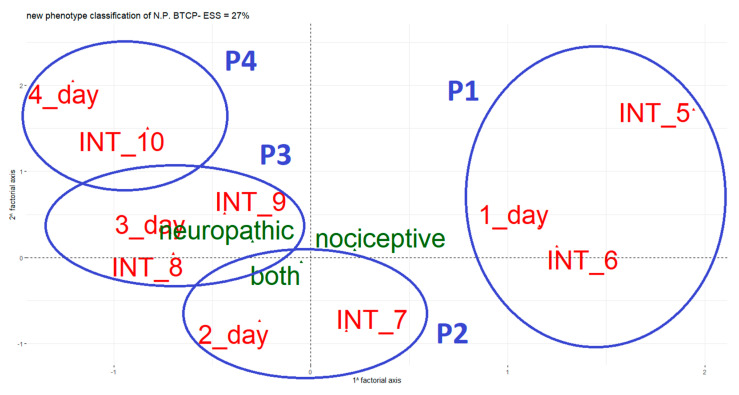
Multiple correspondence analysis (MCA plot). Identification of 4 groups (phenotypes, P) of breakthrough cancer pain (BTcP) from P1 to P4 by combining the variables BTcP intensity (INT) score (INT_5; INT_6; INT_7; INT_8; INT_9; INT_10) and number of BTcP episodes per day (1_day; 2_day; 3_day; 4_day) with the categorical variable BTcP type (nociceptive, neuropathic, both).

**Figure 3 cancers-13-04018-f003:**
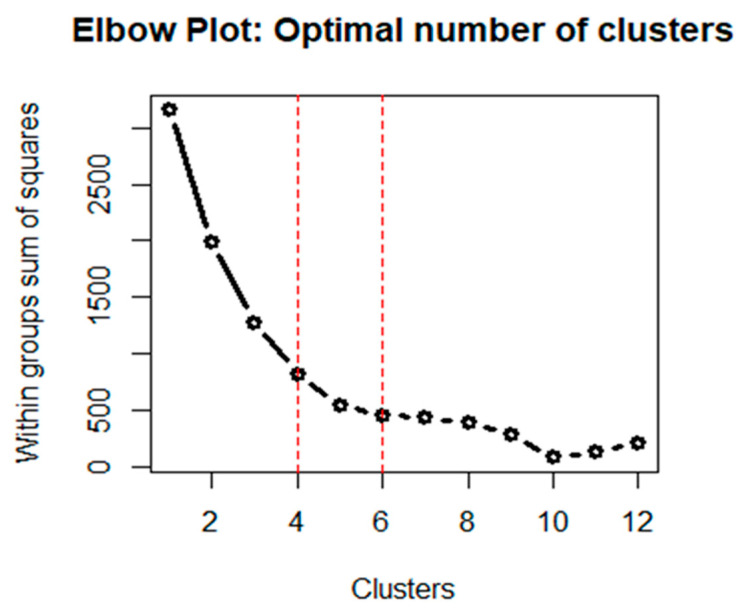
Elbow plot. An automatic procedure recognizes the 6 theoretical clusters groups (within groups sum of squares, WSS6 = 509.13). The k-means method suggests that a good representation of clusters is given from 4 to 10 groups. By using a parsimonious point of view, 4 clusters (phenotypes) were selected (WSS4 = 838.93). According to cluster estimation, the agreement between induced classification and automatic classification was 31.6%.

**Figure 4 cancers-13-04018-f004:**
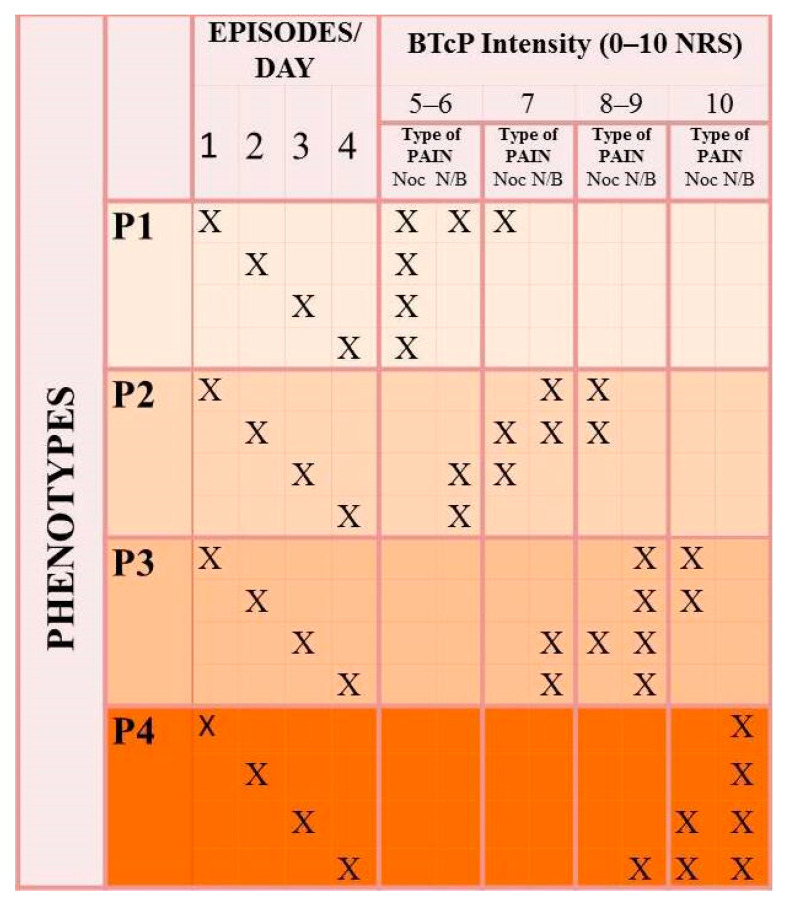
Phenotypes of non-predictable breakthrough cancer pain. Four phenotypes (from P1 to P4) are described according to the pain intensity, the number of episodes a day, and the type of pain. Abbreviations: NRS—numerical rating scale; Noc—nociceptive; N/B—neuropathic or both.

**Figure 5 cancers-13-04018-f005:**
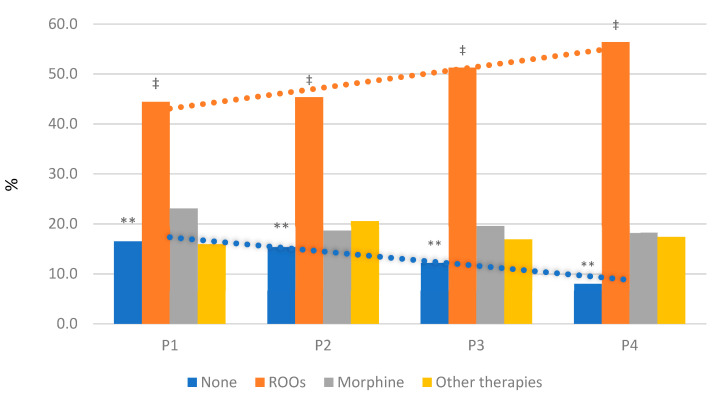
Distribution of therapies according to phenotypes. The NP-BTcP management differs between phenotypes. Orange trend indicates significant increase of ROOs from P1 to P4 (‡ *p* = 0.008); blue trend shows significant decrease of none therapy from P1 to P4 (** *p* = 0.003). Abbreviations: NP-BTcP—non-predictable breakthrough cancer pain; ROOs—rapid-onset opioids.

**Table 1 cancers-13-04018-t001:** Descriptive Statistics.

	*n* (Total = 2671)	%
Gender		
Male	1465	54.8
Female	1206	45.2
Age		
<55	589	22.1
55–64	696	26.1
65–74	810	30.3
≥75	576	21.6
Setting		
Inpatients	1015	36.0
Outpatients/DH	1250	46.8
Hospice/Home palliative care	406	15.2
Primary Tumor		
Lung	665	24.9
Gastrointestinal	470	17.6
Breast	278	10.4
Pancreas	237	8.9
Uro-gynecological	401	15.0
Other	620	23.2
*n*° of BTcP		
1	791	29.6
2	1082	40.5
3	621	23.2
4	177	6.6
Intensity of BTcP		
5	118	4.4
6	394	14.8
7	840	31.4
8	849	31.8
9	277	10.4
10	193	7.2
Type of BtcP Pain		
Nociceptive	672	25.2
Neurophatic	214	8.0
Both types	1785	66.8
Onset		
≤10 min	1806	67.6
>10 min	865	32.4
BTcP therapy		
None	366	13.8
ROOs ^	1302	48.7
Morphine °	545	20.4
Other	458	17.1

Abbreviations: RT—radiotherapy; DH—day hospital; ROOs—rapid-onset opioids. Legend: ^ ROOs include: OTFC—oral transmucosal fentanyl citrate; FBT—fentanyl buccal tablet; FBST—sublingual fentanyl; FPNS—fentanyl pectin nasal spray. ° Morphine given by the oral, subcutaneous, intramuscular, or intravenous route.

**Table 2 cancers-13-04018-t002:** Univariate analysis.

	Phenotypes				*p*-Value *
Characteristics	1 (*n* = 394)	2 (*n* = 962)	3 (*n* = 1079)	4 (*n* = 236)	
Age					0.055
<55	68 (17.3%)	220 (22.9%)	244 (22.6%)	57 (24.2%)	
55–64	96 (24.4%)	229 (23.8%)	305 (28.3%)	66 (26.1%)	
65–74	128 (32.5%)	301 (31.3%)	313 (29.0%)	68 (28.8%)	
≥75	102 (25.9%)	212 (22.0%)	217 (20.1%)	45 (19.1%)	
Gender					0.5
Female	177 (44.9%)	452 (47%)	476 (44.1%)	101 (42.8%)	
Male	217 (55.1%)	510 (53%)	603 (55.9%)	135 (57.2%)	
Onset					**<0.001**
≤10 min	247 (62.7%)	552 (57.4%)	832 (77.1%)	175 (74.2%)	
>10 min	147 (37.3%)	410 (42.6%)	247 (22.9%)	61 (25.8%)	
Setting					**<0.001**
Inpatients	167 (42.2%)	308 (32.0%)	437 (40.5%)	103 (43.6%)	
Outpatients/DH	122 (31.0%)	487 (50.6%)	539 (50.0%)	102 (43.2%)	
Hospice/Home palliative care	105 (26.6%)	167 (17.4%)	103 (9.5%)	31 (13.1%)	
Cancer type					**<0.001**
Lung	67 (17.0%)	232 (24.1%)	298 (27.6%)	68 (28.8%)	
Gastric	104 (26.4%)	173 (18.0%)	161 (14.9%)	31 (13.1)	
Breast	41 (10.4%)	110 (11.4%)	104 (9.6%)	23 (9.7%)	
Pancreatic	38 (9.6%)	80 (8.3%)	101 (9.4%)	18 (7.6%)	
Uro-gynecological	42 (10.7%)	146 (15.2%)	172 (15.9%)	41 (17.4%)	
All other cancers	102 (25.9%)	407 (42.5%)	459 (42.5%)	101 (42.8%)	
Therapy					**0.008**
None	65 (16.5%)	150 (15.6%)	132 (12.2%)	19 (8.1%)	
ROOs	175 (44.4%)	441 (45.8%)	553 (51.3%)	133 (56.4%)	
Morphine °	91 (23.1%)	182 (18.9%)	211 (19.6%)	43 (18.2%)	
Other therapies	63 (16%)	200 (20.8%)	183 (17%)	41 (17.4%)	

* Chi-square test. Abbreviations: RT—radiotherapy; DH—day hospital; ROOs—rapid-onset opioids. Legend: ° Orally morphine sulfate, morphine hydrochloride subcutaneous or intravenous.

## Data Availability

The datasets generated during/and or analyzed during the current study are available from the corresponding author on request.

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
