# Peer review of "Multidimensional Statistical Technique for Interpreting the Spontaneous Breakthrough Cancer Pain Phenomenon. A Secondary Analysis from the IOPS-MS Study"

_cancers, 2021, doi:10.3390/cancers13164018_

Round 1
Reviewer 1 Report
The article describes a statistical technique to classify breakthrough cancer pain, characterizes 4 groups and gives suggestions for treatment. The article is important, since its findings may lead to a better characterisation and managanet of breaktrhough cancer pain.
1. There is something unclear to me: on the one hand, in the Materials and methods, section statistical analysis, it is written in the first paragraph that BTcP intensity score, number of BTCP episodes per day and duration of BTCP are continuous variables. On the other hand, these variables where subject to MCA, which requires variables to be not continuous (otherwise we have principal components analysis). The use of the Pearson correlation on the one hand (requiring continuous variables) and MCA on the other hand (requiring non-continuous variables) sounds contradictory to me and requires more explanation.
2. Section 3. results: first sentence (This section may be... can be drawn) should be omitted.
3. Figure 2: this graph is unclear. There are too much codes standing there that are not explained, like INT_5, INT_6, INT_8, 1_day, and so on. I believe that the different variables have been split in different binary variables. If this is the case, this should be explained.
4. The cluster analysis, represented in the results with the elbow plot, has not been described in the materials and methods.
5. In the discussion: third paragraph: ... and a large size of the sample: this should be rewritten. Did the authors mean 'a large sample size' ?
Author Response
Dear Editor,
Thank you for your letter and the opportunity to revise our paper entitled:
Multidimensional statistical technique for interpreting the spontaneous breakthrough cancer pain phenomenon. A secondary analysis from the IOPS-MS study
The suggestions offered by the reviewers have been immensely helpful, and we also appreciate their insightful comments on revising the methods of the paper.
We have included the reviewer comments immediately after this letter and responded to them individually, indicating exactly how we addressed each concern or problem and describing the changes we have made. The revisions have been approved by all four authors and the changes are marked in the paper as you requested.
We appreciate your valuable time for editing our manuscript.
We hope this revised version is suitable for publication in your prestigious anesthesia journal.
Reply to Reviewers’ comments.
Reviewer n.1
The article describes a statistical technique to classify breakthrough cancer pain, characterizes 4 groups and gives suggestions for treatment. The article is important, since its findings may lead to a better characterisation and managanet of breaktrhough cancer pain.
Reply: We thank the reviewer for the appreciation of the paper.
- There is something unclear to me: on the one hand, in the Materials and methods, section statistical analysis, it is written in the first paragraph that BTcP intensity score, number of BTCP episodes per day and duration of BTCP are continuous variables. On the other hand, these variables where subject to MCA, which requires variables to be not continuous (otherwise we have principal components analysis). The use of the Pearson correlation on the one hand (requiring continuous variables) and MCA on the other hand (requiring non-continuous variables) sounds contradictory to me and requires more explanation.
Reply: We thank the reviewer for this question. The variables: “BTcP intensity score” and “number of BTCP episodes per day” are quantitative discrete variables (not-continuous) for which the values are countable and have a finite number of possibilities; on the other hand, the variable “duration of BTCP” is a quantitative continuous variable for which is not countable and has an infinite number of possibilities. Thus, the Pearson correlation was used at first to assess the correlation among the three quantitative variables; once the statistically significant association was found between the “BTcP intensity score” and “number of BTCP episodes per day” that assume values from 5 to 10 and from 1 to 4 respectively, we considered these two quantitative discrete variables as categorical.
Please note that we modified this statement in the Materials and Methods Section.
- Section 3. results: first sentence (This section may be... can be drawn) should be omitted.
Reply: we omitted the first sentence in the Results Section as the reviewer suggested.
- Figure 2: this graph is unclear. There are too much codes standing there that are not explained, like INT_5, INT_6, INT_8, 1_day, and so on. I believe that the different variables have been split in different binary variables. If this is the case, this should be explained.
Reply: we added in the legend of the Figure 2 the exact correspondence between labels of the variables and the codes in the graph.
- The cluster analysis, represented in the results with the elbow plot, has not been described in the materials and methods.
Reply: Thank you for your suggestion. This helpful information was added (see the text_Methods). The cluster analysis is a technique which reduces the number of observations by classifying them into homogeneous clusters, identifying the groups without previously knowing group membership or the number of possible groups. The Elbow-plot (Figure 3) is the a graphical tool that helps to choose the best number of clusters for HCPC.
- In the discussion: third paragraph: ... and a large size of the sample: this should be rewritten. Did the authors mean 'a large sample size' ?
Reply: Thank you for your comment we modified the sentence according to your suggestion.

Reviewer 2 Report
Thank you for permitting me to review this manuscript
here are my comments
Please insert somewhere in the introduction the definition of cancer pain and differentiate it with other form of chronic pain with reference
Since the ethical approval was of 2013 for another purpose , a new IRB approving the use of the survey for this new work is necessary at least in some countries. please check with your IRB if this new work need a new approval.
Please explain why patients having more than 4 episodes of pain were excluded , provide reference if available.
Please explain how you describe 4 CATEGORIES
1 NOCICEPTIVE
2 NEUROPATHIC
3 MIXT if the mixt is absent we go back to 1 or 2 n ot a 4 cathegory
which is displayed in the table page 5 at this stage for me there is only 3 categories
It appears to me the 4 category was designed later after analysing result and using another stattistical method. why 4 category was choosed specifically ? Please make it clear and concise., it is somehow difficult to navigate between real data and statistical methods adviser
It would be better to describe very briefly the 4 phenotypes by a few words
This need a clear explanation as right now it is confusing
Please try to explain in the discussion why the worst case (3-4) needed more ROO and less none therapy this is also somehow a confusing message
I hope this will help
The authors should clearly state in the conclusion that this classification need validation by a prospective study and should not be used before validation.
Author Response
Dear Editor,
Thank you for your letter and the opportunity to revise our paper entitled:
Multidimensional statistical technique for interpreting the spontaneous breakthrough cancer pain phenomenon. A secondary analysis from the IOPS-MS study
The suggestions offered by the reviewers have been immensely helpful, and we also appreciate their insightful comments on revising the methods of the paper.
We have included the reviewer comments immediately after this letter and responded to them individually, indicating exactly how we addressed each concern or problem and describing the changes we have made. The revisions have been approved by all four authors and the changes are marked in the paper as you requested.
We appreciate your valuable time for editing our manuscript.
We hope this revised version is suitable for publication in your prestigious anesthesia journal.
Reply to Reviewers’ comments.
Reviewer n.2
Please insert somewhere in the introduction the definition of cancer pain and differentiate it with other form of chronic pain with reference
Reply: We included the IASP definition (and reference)… ‘According to the IASP (International Association for the Study of Pain), chronic cancer pain is defined as chronic pain caused by the primary cancer itself or metastases or its treatment (see text)’.
Since the ethical approval was of 2013 for another purpose , a new IRB approving the use of the survey for this new work is necessary at least in some countries. please check with your IRB if this new work need a new approval.
The primary purpose of the “original” Study was to monitor and characterize the BTcP in order to identify the type of cancer where it is most present and in which stage of the tumor disease occurs. Moreover, the primary objective was to identify the triggering events of the BTcP and if there was a relationship with the background pain and its therapy or if it is associated with other relevant clinical phenomena. So, this secondary analysis of a specific subgroup of patients with unpredictable BTcP still fulfil the original objective, as reported in the manuscript and new IRB approving the use of the survey for this new work was not necessary.
Please explain why patients having more than 4 episodes of pain were excluded , provide reference if available.
Reply: Thank you for your comment. We agree that there is no unanimous consensus on the clinical features to define BTcP. To consider that baseline pain is adequately controlled, some authors assume that the average intensity of pain must be less than four on a verbal numerical rating scale (VNRS) or visual analogue scale (VAS) from 0 to 10, and the maximum number of episodes of BTcP should be three or four per day. On the other hand, there are some typical episodes that are triggered by several factors, for example, incident pain due to bone metastases, which can occur more frequently (>4/day episodes). Most of these BTcP episodes are elicited by physical activity (e.g., in the presence of bone metastases). Nevertheless, despite this uncertainty mainly concerns incident pain, in spontaneous BTcP a number of over 4 episodes per day is very indicative of a not well treated background pain. We have dealt with this aspect in several studies on BTcP (see papers on the topic of Mercadante S.). We included a reference.
Please explain how you describe 4 CATEGORIES
1 NOCICEPTIVE
2 NEUROPATHIC
3 MIXT if the mixt is absent we go back to 1 or 2 n ot a 4 cathegory
which is displayed in the table page 5 at this stage for me there is only 3 categories
Reply: Thanks for your comment. The pain categories are “nociceptive”, “neuropathic” and “both” and not “mixed” which was a taping error and could get confusing.
It appears to me the 4 category was designed later after analysing result and using another stattistical method. why 4 category was choosed specifically ? Please make it clear and concise., it is somehow difficult to navigate between real data and statistical methods adviser
Reply: The Multiple Correspondence Analysis (MCA) determined the identification of groups (termed as ‘Phenotypes’), by MCA-plot we recognized 4 groups (phenotypes). Then a confirmative hierarchical clustering principal components analysis (HCPC) was implemented with a parsimonious criterion to compare the theoretical findings through an automatic classification method.
It would be better to describe very briefly the 4 phenotypes by a few words. This need a clear explanation as right now it is confusing
Reply. The 4 Phenotypes represent the classification levels which summarize health conditions of a patient suffering of BTcP, in particular they are the combination of the variables: BTcP intensity score”, “number of BTCP episodes per day” and “Type of BTcP pain” as reported in Figure 4.
Please try to explain in the discussion why the worst case (3-4) needed more ROO and less none therapy this is also somehow a confusing message. I hope this will help
Reply. Thank you for your comments; we agree that these data can induce confusing messages. Nevertheless, we limited to observe the characteristics of our sample according to their phenotypes. Other studies are needed to confirm/validate these results (see below). Again, this result underlies that ‘Despite the guidelines on the topic, proper cancer pain treatment remains a paramount unmet medical need for cancer patients’. Indeed, as we reported, ‘guidelines on cancer pain treatment do not refer to the severity of BTcP and provide different guidance regarding its treatment’.
The authors should clearly state in the conclusion that this classification need validation by a prospective study and should not be used before validation.
Reply. We really appreciate your evaluable statement. We added in the conclusion the following sentence:
The aim is to provide tailored management of the NP-BTcP through designing prospective study for the validation of this multidimensional statistical approach.
This point was also addressed in the abstract.

Round 2
Reviewer 2 Report
The authors has responded to most of my queries. Here are the points in which the authors did not totally answered the queries
Please insert somewhere in the introduction the definition of cancer pain and differentiate it with other form of chronic pain with reference
In their response there is no a real differentiation with other chronic pain
The ethical approval is 8 years old and the authors say that nothing has changed
I still don't understand why there is 4 categories I only see 3 kind of pain
Author Response
Reply to Reviewer’s comments.
Reviewer n.2
In their response there is no a real differentiation with other chronic pain
Reply: Thank you for your comment. We clarify the differentiation between chronic cancer pain with other chronic pain and we added a new reference as the reviewer suggested. (see text)’.
The ethical approval is 8 years old and the authors say that nothing has changed
Reply: We confirm that our Ethical Commette (EC) did not ask for further approval this secondary analysis and new IRB for this new work was not necessary.
I still don't understand why there is 4 categories I only see 3 kind of pain
Reply: Sorry for the typo, the pain categories are 3: “nociceptive”, “neuropathic” and “both”.
